# “It’s Not Important”—The Social Constructing of Poor Physical Health as ‘Normal’ among People with Schizophrenia: Integrated Findings from an Ethnographic Study

**DOI:** 10.3390/ijerph20126133

**Published:** 2023-06-15

**Authors:** Birgitte Lerbæk, Rikke Jørgensen, Andrea McCloughen

**Affiliations:** 1Unit for Psychiatric Research, Psychiatry, Aalborg University Hospital, 9000 Aalborg, Denmark; 2Department of Clinical Medicine, Aalborg University, 9000 Aalborg, Denmark; 3Susan Wakil School of Nursing and Midwifery, Faculty of Medicine and Health, The University of Sydney, Sydney, NSW 2050, Australia

**Keywords:** ethnography, schizophrenia, self-management, social interaction, healthcare delivery, qualitative research

## Abstract

People with schizophrenia have shortened life expectancy partly due to physical ill health. Management of coexisting mental and physical health issues is complex, and knowledge in the field is lacking. This study investigated how physical health was managed among people with schizophrenia, by integrating findings from three separate analyses conducted in an ethnographic study. Qualitative data generation methods were used; 505 h of field work were undertaken among nine participants with schizophrenia and 27 mental healthcare professionals were interviewed using a semi-structured interview approach. Three separate analyses were conducted using thematic and discourse analysis. Progressive focusing was used to integrate findings. Across the mental health care contexts that were part of this research, managing physical health was characterised by a lack of recognition of the seriousness of physical health issues as part of everyday life among people with schizophrenia. Poor physical health was accounted for as being “not of importance” by both mental health care professionals and the participants experiencing physical health issues. The integrated findings offer new insights about the social co-construction of poor physical health as something normal. At the individual level, this shared understanding by people with schizophrenia and healthcare professionals contributed to sustaining inexpedient management strategies of “modifying” behaviour or “retreating” from everyday life when physical health issues were experienced.

## 1. Introduction

It is well-established that people with mental illness such as schizophrenia are at high risk of developing physical comorbidities and have shortened life expectancy when compared to the general population [1,2,3,4]. Within the traditions of medicine and psychiatry, schizophrenia is characterised as a severe mental illness (SMI) associated with severe cognitive deficits [5]. Individual experiences of mental health problems and the complexities associated with these problems being part of everyday life, are not well-reflected in medicalised language that relies on terms such as ‘schizophrenia’ or ‘severe mental illness’ [6,7]. However, the terms are used here as they reflect the language of the sites where our study took place and are commonplace in the broader literature. We have aimed to keep in mind the unique lived experiences of people affected by mental health problems.

Existing research into the provision of physical health care to people with schizophrenia reports inadequate general physical health care [8] and inadequate treatment of physical illnesses such as diabetes and heart failure [9,10,11]. People with schizophrenia may experience difficulties with executive functions such as planning and performing everyday activities such as preparing meals and cleaning ones’ home, and may require substantial support from others to maintain stability in various aspects of everyday life [5,12]. Recent research found a significant association between poor cardiovascular health outcomes and cognitive deficits among people with schizophrenia [13]. While knowledge about the management of coexisting schizophrenia and physical ill health is limited [14], self-management of physical and mental comorbidity has been described as being impacted by barriers at three domains of individual, family and community, and provider and healthcare system levels [15]. These barriers include social isolation and stigma (individual level), lack of support from relatives and friends (family and community level), and poor communication with services, fragile relationships, and fragmentation of health care services (provider and healthcare system levels) [15]. In addition, provision of physical healthcare to people with SMI include barriers at a structural or macro level, such as geographical distances to services, challenges associated with attending many appointments in a fragmented system, and financial constraints [16]. For more than two decades, the organisation of contemporary healthcare systems has been influenced by traditional body–mind dualist thinking [17]. This duality promotes siloed organisational structures in health care and fragmentation of services across healthcare systems [18], which create barriers to the promotion of physical health, detection and treatment of coexisting mental and physical illnesses. Consequently, in some mental health care settings the management of physical health issues is marginalised as a strategy to safeguard tenuous clinical relationships or to manage dilemmas embedded in health promoting activities [19,20].

Roebroek et al. [21] propose that special attention is required for improving physical health in psychosis care [21]. These authors suggest that a recovery-oriented view and well-coordinated collaboration between clinicians and general practitioners, together with shared decisions about which care needs to treat, can improve treatment delivery [21]. This approach encompasses person-centred care, integrated care, and models of care involving close monitoring of physical health [3,16], which have all been suggested as possible solutions to the dilemma of providing sufficient support to people with SMI to manage physical health issues. However, despite these suggested solutions, the management and prevention of physical health issues among people with SMI remains suboptimal and continues to gain clinical and political attention [22]. The collective impacts of individual, community, and health system barriers highlight the challenges of managing comorbidities and need to be considered in any model of support offered to people experiencing them [15,16].

Further research into what influences (self)management of physical health in the everyday life of people experiencing severe mental illness is needed to gain greater understanding of the complexities that come into play in these social contexts. This paper reports the integrated findings from an ethnographic study that aimed to explore aspects of everyday life among a group of people with schizophrenia to gain insight into how physical health issues were managed.

## 2. Materials and Methods

### 2.1. Design

This research was designed as an ethnographic study [23], and the theoretical perspective of social constructionism was applied; hence the study built on the assumption that our understanding of the world is continually shaped through social interactions. As part of this, certain patterns of social behaviour are sustained while others are excluded. Implications are related to how we engage with people around us and what we perceive as expected behaviour in everyday life situations [24].

### 2.2. Materials

We wanted to provide insight into management of physical health issues in everyday life among people with schizophrenia. To gain this knowledge, we sought insight into the perspectives of a group of people diagnosed with schizophrenia (*n* = 9), and of mental health care professionals (MHCPs) (*n* = 27) involved in everyday care and treatment in the home environment of the participants with schizophrenia. Participants with schizophrenia were recruited using purposeful sampling [25]. This approach allowed us to recruit participants with the specific characteristic of having the diagnosis of schizophrenia. Mental health care professionals in the included settings assisted in the recruitment. The included settings were one outpatient clinic for younger people with newly diagnosed schizophrenia and two mental health care residential facilities. The residential facilities were organized under the regional health authorities’ special sector and targeted people with severe physical and/or mental health issues. The residents had access to staff at all hours.

The first author was the primary investigator and was responsible for generation of data materials. In total, 505 h of field work was conducted among the participants diagnosed with schizophrenia in settings that represented their everyday life. The participants either lived in their own home or in shared apartments in the community (*n* = 5) or at mental health residential facilities (*n* = 4) in the North Denmark Region. The age of these nine participants ranged from 21–70 years (median: 30; IQR: 25-55). All but two were men. Eight were single and none had children. The data materials generated through field work included fieldnotes, semi-structured interviews, and audio-recorded informal conversations with participants with schizophrenia. In addition, this ethnographic study included data from three focus groups conducted among MHCPs (*n* = 22) [26] and semi-structured, individual interviews with key informants (also MHCPs) (*n* = 5) [27]. These participants represented different occupational groups such as registered nurse, social and health care assistant, nursing assistant, social support worker, psychologist, and occupational therapist. They were recruited from the outpatient clinic or the residential facilities as associated with the participants with schizophrenia. They reported median age of 51 years (IQR: 36.5–55) and current employment ranged from 5 months to 38 years. In total, six of these participants were men.

Detailed information on the study participants has been reported elsewhere [26,27,28].

Figure 1 displays the exploratory sequences of the study design.

**Figure 1 ijerph-20-06133-f001:**
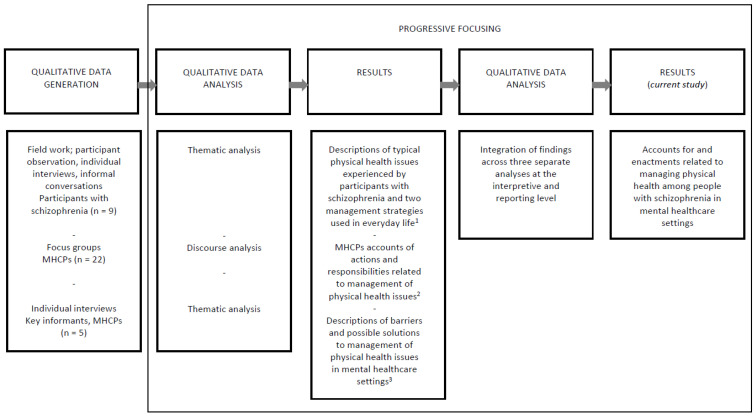
Exploratory sequences of the study design.

^1^ Lerbæk et al. (2021); ”Modifying” or ”Retreating”—Self-management of physical health among a group of people with schizophrenia. An ethnographic study from Denmark. In: International Journal of Mental Health Nursing; 30:6, pp. 1575–1587 [28], ^2^ Lerbæk et al. (2019); Mental health care professionals’ accounts of actions and responsibilities related to managing physical health among people with severe mental illness. In: Archives of Psychiatric Nursing; 33, pp. 174–181 [27], ^3^ Lerbæk et al. (2021); Barriers and possible solutions to providing physical health care in mental health care: A qualitative study of key informants’ perspectives. In: Issues in Mental Health Nursing; 42:5, pp. 463–472 [26].

### 2.3. Analytical Strategy

The overall process of analysis entailed progressive focussing [23], which can be explained as a process of continuously focusing the scope of the analysis, moving from a descriptive level towards a more interpretative and theorising level. Initial management of the data materials entailed conducting three separate analyses: (1) A thematic analysis [29], which led to the description of debilitating physical health issues experienced by participants with schizophrenia (*n* = 9) and the strategies they used to manage them in everyday life. These findings were central to the ethnographic study and have been reported elsewhere [28]; (2) A discourse analysis [30,31] of focus group discussions with MHCPs (*n* = 22) [26]; and (3) A thematic analysis of interviews with key informants (*n* = 5) [27]. The latter two analyses provided contextual depth to explanatory ideas about how social context and social interactions played a role in management of physical health issues among the participants with schizophrenia.

Table 1 provides an overview of the data material and analyses. 

The next step in the process of progressive focusing [23] was concentrated on integrating the findings of the three separate analyses at an interpreting and reporting level. The results of this process are reported in this paper. Integration was conducted to reach in-depth understanding of the management of physical health as social behaviour, which was continuously shaped through social interactions taking place in everyday life of the participants with schizophrenia. The integration technique included revisiting the findings produced in the three separate analyses and was guided by specific key points focused on identifying ‘*what was said*?’ and ‘*what was done*?’ about management of physical health issues across each analysis. This was an iterative process involving movement back and forth between the three separate analysis, the data material, and the integration of findings.

In this process two themes were derived describing how management of physical health issues as part of everyday life among people with schizophrenia was accounted for and enacted in the study context. Hence, the scope was narrowed towards capturing the social construction of physical health among participants with schizophrenia and MHCPs, and how a joint understanding of physical health was sustained through social interactions. Table 2 displays themes and examples of the integrated interpretation.

Text presented in italics and single inverted commas are direct quotes or extracts from the data material.

### 2.4. Ethics

All participants received written and oral information about the study prior to providing written or oral informed consent. The Danish Data Protection Agency was notified about the study and regulations regarding data management were followed. Approval from the research ethics board was not required according to Danish regulation code.

## 3. Results

The process of integrating findings from the three separate analyses led to the identification of two themes: (1) de-prioritizing physical health issues in everyday life and practice, and (2) limited management of physical health issues in everyday life settings (see Table 2). Together these two themes illustrate how poor physical health was constructed as normal for people with schizophrenia—a shared understanding between those with schizophrenia and healthcare professionals that was sustained through ongoing social interactions and by certain actions that took place within the explored contexts.

At an overall level, these new findings point to the presence of a general attitude in the included mental health care contexts, where the experienced physical health issues were deprioritized and not recognized as serious enough to require attention.

### 3.1. De-Prioritizing Physical Health Issues in Everyday Life and Practice

This theme describes how a lack of management of physical health issues in everyday life was accounted for by participants with schizophrenia and MHCPs. Exploring the experiences of physical health issues in everyday life among the participants with schizophrenia revealed the presence of various physical health issues. It also revealed how the social context of participants’ everyday life was characterized by a lack of interactions with others about their prevailing and ongoing physical health issues.

The prevailing physical health issues were characterized as either ongoing symptoms of physical ill health that seemed to worsen over time, or repeated, discrete episodes of symptoms of severe physical ill health. The physical health issues experienced by participants with schizophrenia included both specific chronic conditions such as diabetes, chronic obstructive pulmonary disease (COPD), and arthritis, and symptoms of undefined health issues such as stomachache, diarrhea, vomiting etc. When speaking about how the issues impacted their everyday lives, participants with schizophrenia described ‘*not doing well at all*’ and these issues making it ‘*really, really hard, you know, to get out the door and stuff like that*’. In this way, the prevailing physical health issues were limiting to their activities of daily living and participation in the world.

Despite the presence of these physical health issues and the impact they had on their everyday life, the participants with schizophrenia generally described themselves as healthy and were often reluctant to speak about physical health issues or illness during the study. The participants with schizophrenia who experienced physical health issues most often claimed that the symptoms were ‘*not of importance*’ or ‘*not something I worry about*’, even though they impacted greatly on their ability to function in everyday life. Some was convinced that the symptoms they experienced were “*just an illusion*” and “*something my mind is making up*”.

MHCPs’ accounts of physical health issues among mental healthcare users depicted how they believed people with schizophrenia to have poorer physical health than the general population. One MHCP said:

“*I would say, from an objective perspective, that the health of our patients is generally poor. They are smokers, many don’t eat breakfast […] Generally, the health isn’t good. And those, who I see, who for example has insulin-requiring diabetes, I have had a couple of patients who has that, it’s [diabetes] managed poorly*”.(MHCP, Focus group 3)

Having poorer health was by most of the MHCPs associated with living everyday lives that exposed people with schizophrenia to multiple risk factors due to a lifestyle that was characterized as ‘*unhealthy*’. In addition to smoking and poor eating habits an ‘*unhealthy lifestyle*’ was illustrated to include inactivity, substance use and irregular sleep.

According to MHCPs, another aspect related to the physical health state of people with SMI in general was that they were probably undertreated for physical illnesses.

Despite acknowledging that people with SMI experienced poor physical health, it seemed that deprioritizing management of physical health was an accepted part of everyday life and professional practices. Several MHCPs reflected on the reasons for this and described a general lack of recognition of the seriousness of physical health issues which occurred among their service users. A key informant illustrated this by saying:

“*The staff have gotten used to this miserable health status actually, and well they [people with SMI] have these four rotten pegs in the mouth for example and we don’t really get anything done about it*”.(Key informant 4)

In everyday life, a lack of recognition of the seriousness of the prevailing physical health issues was verified by the very limited social interactions focused on management of physical health that occurred between MHCPs and participants with schizophrenia. In addition, this was reinforced by the common understanding that resonated in accounts by participants with schizophrenia and MHCPs that physical health issues were not a primary concern or a priority in everyday life or in everyday work practices. Physical health issues—whether present as ongoing symptoms indicative of physical ill health or as a specified diagnosis—typically came second to managing mental health issues.

That physical health was given low priority was reflected in the accounts of participants with schizophrenia who would state how ‘*mental health is the thing I need to concentrate on’* and that physical health issues were ‘*not something to waste time talking about*’. In line with this way of deprioritizing physical health issues, accounts from the MHCPs explicitly showed how dealing with physical health issues came second to managing mental health. They described that dealing with aspects of physical health was simply ‘*not a priority’* and that initiatives to manage physical health among people with SMI typically relied on a few MHCPs doing ‘*something extra*’.

Accounts from across the data material illustrated that physical health issues only became a priority to MHCPs when something was “*visibly wrong*” or if the person in question was “*acting different than normal*”. One MHCP explained: 

“*But that’s where we really go in and prioritize. You know, when it’s diabetes or something that can go wrong, that can become a danger to life or something*”.(MHCP, Focus group 2)

For MHCPs their responses to observed changes in physical health were often underpinned by questioning whether to get involved with managing physical health issues, as one described: 

“*Many times, it is really difficult and hard work. And many times, we lose, because the mental health aspect is also involved. Or we don’t lose, the person in question loses, or we have to give up*”.(MHCP, Focus group 2)

Dealing with people with persistent and serious mental illness seemed to constitute a barrier to managing physical health issues. MHCPs accounts depicted how poor mental health could become justification for decision-making when MHCPs decided not to intervene if ’risky’ lifestyle behaviour or potentially damaging physical health conditions were observed among those with SMI. They described how dealing with mental health was the first priority in care provision in these settings. Some added a notion of having to choose one or the other area to focus on: 

“*It really is a dilemma, because if you work with one thing then you put pressure on the other. It really is one or the other. It’s really difficult to reach a point where you can work with both*”.(MHCP, Focus group 2)

Consequently, MHCPs justified their choice to focus their efforts on mental health issues.

Overall, these findings point towards how physical ill health among those diagnosed with persistent and severe mental illness that had become normalized to both those with schizophrenia and the MHCPs. This to the extent that higher levels of poor physical health were accepted as part of their conditions and was depicted as not a priority in these everyday life settings.

### 3.2. Limited Management of Physical Health Issues in Everyday Life Settings

This theme describes how a shared understanding of poor physical health as normal for people with schizophrenia, was sustained in everyday life situations. The theme demonstrates how lack of management of physical health issues was enacted by participants with schizophrenia and MHCPs and how limited everyday practices were explained or justified. This includes examples of actions taken as well as situations where no action was taken to manage actual physical health issues.

Doing field work in these everyday life settings provided insight into some of the ways physical health issues impacted on everyday life. Typically, the participants with schizophrenia attempted to manage their physical health issues by either modifying behaviour in everyday life to avoid discomfort of ongoing ill physical health or retreating from everyday life to recover from discrete episodes of symptoms of severe physical ill health. To those modifying their behaviour to avoid discomfort, the physical health issues gradually increased, causing increased limitations in everyday life. This included increased social isolation and numerous worries and speculations about how to avoid getting in situations that would cause discomfort. Those who retreated from everyday life to recover also faced increased social isolation as they would lie in bed for several days or weeks “*to just go through it to get better”*. Some also described how “*just lying in bed for days*” could have negative impact on their mental health state. Retreating as a way to manage physical health issues impacted on their ability to fulfill even basic human needs, as it typically resulted in decreased ability to manage self-care and neglected personal hygiene. Some would have *‘trouble going out and getting groceries*’ and consequently ended up eating ‘*whatever was in the home’*. During such times, some experienced acute worsening of their physical health state and risked serious complications to coexisting chronic conditions such as diabetes. Poor diet and lack of ability to manage blood glucose levels ended up causing acute admissions to the hospital.

Both strategies used by the participants with schizophrenia to manage physical health issues were inexpedient, as no meaningful improvement in their physical health was reached. When asked about these management strategies, the participants often replied that they chose to modify behaviour or retreat to recover as ‘*it was the easiest’* but also added that ‘*it’s not really as if it has helped*’.

Even though these strategies did not facilitate positive change in the participants’ physical health, both strategies seemed to induce a level of control and predictability to the situation and were repeatedly demonstrated by the participants throughout the study. During the time of the study, the management strategies were not questioned by the MHCPs close to them, and in general, everyday life of the participants with schizophrenia was characterized by limited social interactions with other people. For most of them, the MHCPs represented central social actors in their lives, however, based on the field work conducted as part of the study, it was apparent that to a great extent interactions lacked focus on physical health issues. The interactions that played out between participants with schizophrenia and MHCPs were mainly concerned with managing symptoms, medication and other issues related to mental health that caused worries in everyday life.

MHCPs’ accounts depicted how physical health issues among the users with severe mental illness tended to be overlooked due to prominent mental health issues. These statements included descriptions of how people with schizophrenia were victims of ongoing stigmatization in both general healthcare and mental healthcare settings. Living everyday lives that were dominated by severity and persistence of their mental health diagnosis was a factor that impacted on people with schizophrenia having a poor physical health state.

These integrated findings gave insight into MHCPs beliefs and awareness of the physical health issues that the mental health users were facing and how they were involved in managing these. It seemed that the beliefs and attitudes held by MHCPs contributed to their limited involvement in the management of physical health concerns for people with schizophrenia. Despite acknowledging that people with schizophrenia were often stigmatized, they lacked an understanding of their own behaviour and how it contributed to this stigma, i.e., blaming people for their poor physical heath or having very limited expectations of people's capability. One MHCP said:

“*Things get stranded in everyday life, because they don’t have the surplus energy to get past that hurdle and get things done due to their [mental health] diagnosis. They don’t really get out the door. And I really think that’s a huge barrier to them becoming a bit healthier*”.(MHCP, Focus group 3)

The challenges experienced by MHCPs when working with people experiencing symptoms associated with SMI, were used as an argument to de-prioritize physical health issues in their everyday work practices. MHCPs described paranoid delusions, hallucinations, and cognitive deficits as problematic and explained how they would sometimes choose not to act on physical health issues, because previous attempts to intervene had been unfruitful. One MHCP explained; “*[SMI] is just difficult to compete with […] because you can’t convince someone with paranoia, that it doesn’t exist*’.

This attitude among MHCPs also applied to limited management of some specific physical health diagnosis such as diabetes: 

‘*Well, with Brian it’s also just difficult to speak with him about his diabetes, if he is standing in the parking lot screaming at the stars, that he is God […] you know, that is just not possible*’.(MHCP, Focus group 1)

Based on attitudes such as the ones expressed here, some MHCPs believed it was pointless to try to reason with a person who was in such a poor mental state and therefore ended up doing nothing about their physical health. Providing extreme cases of how difficult it was to persuade people with SMI to think differently about particular issues, served to underline MHCPs’ arguments and hence justifications for not acting on physical health concerns in some aspects of their everyday practice.

Some MHCPs described how their limited intervention in relation to physical health was also associated with a lack of professional knowledge about the complexities in managing coexisting physical and mental health issues. An issue related to this was that “*they actually don’t get to the general practitioner regularly enough”.* This was illustrated in how some who lived with coexisting schizophrenia and chronic conditions such as COPD or diabetes would not get to annual follow-up appointments at their general practitioner, and that the MHCPs responsible for providing support in everyday life settings simply did not know that such follow-ups were expected. In some of these settings, the MHCPs did not have education or training in healthcare, did not realize that this was important.

Some MHCPs described experiences of resistance among their colleagues when they attempted to incorporate physical health strategies into their practice. For example, if they had to do extra tasks during shifts to prepare a healthy snack for residents as part of an initiative to help overweight residents make healthier food choices in everyday life. One of the key informants said: “*I had a lot of colleagues who thought it was bloody annoying*”. Such lack of support from colleagues made promotion of physical health and preventive care a demotivating and ‘*uphill task*’. One said: 

“*I run dead in it sometimes and think “then it might as well bloody not matter. I won’t do anymore. You know, I don’t want to keep slaving at it*”.(Key informant 1)

Interactions between participants with schizophrenia and MHCPs related to managing physical health issues, mainly occurred when the physical health state of those with schizophrenia had deteriorated to the point where they would employ the strategies of modifying behaviour to avoid discomfort or retreating from everyday life to recover. Even in situations where physical ill health in people with SMI was detected by MHCPs, some would stand back and do nothing, thereby silently accepting how poor physical health could impact on and disrupt the person’s everyday life. Examples of these impacts included people with such poor dental status that they were unable to eat proper food, and people with poor mobility and ongoing experiences of pain becoming isolated in their homes. Some MHCPs described how they and their colleagues would act as bystanders and ‘*not really get anything done about it*’.

The integrated findings revealed how the management of prevailing physical health issues in these social contexts resulted in a deadlock situation, as the actions taken from both sides did not result in any improvement in health among the people with schizophrenia. In this way, the social interactions among people with schizophrenia experiencing physical health issues and MHCPS (or the lack of) sustained inexpedient ways of managing physical health as part of everyday life.

## 4. Discussion

This study focused on the integration of findings across three separate analyses to describe management of physical health in everyday life among a group of people with schizophrenia. An important finding was the description of an ongoing social construction of poor physical health among people with schizophrenia as ‘*normal’*, ‘*expected’*, and ‘*not of importance*’. In the context of this study, everyday life situations and work practices resulted in a deadlock where physical health issues and the inexpedient strategies used by participants with schizophrenia to manage them, were continuously sustained, rather than discontinued or challenged by MHCPs. The shared understanding of physical health issues as something ‘*expected’* led to a general lack of recognition of the potential seriousness of these issues.

There is a general sociological understanding that life is organised around routines and that the ‘ordinariness’ of everyday life is socially produced through social interactions [34,35]. Normalization of abnormal conditions is a way of creating a manageable state for ordinary life [34]. When considering the social context of this research, which was illustrated by an everyday life environment in which people with schizophrenia experienced various ongoing physical health issues and daily challenges related to mental and physical health, both the participants with schizophrenia and the MHCPs may have become habituated to these issues as a usual and expected aspect of life. Even though there may be some community and professional outrage about how people with SMI face slowly deteriorating physical health and early death [36,37,38], at the same time there is high potential for this poor physical health to be perceived as a normal part of everyday life for this group. Witnessing (by MHCPs and others) or experiencing (by people with schizophrenia) poor physical health over and over again can become a routine that remains unchallenged, which in turn minimises or blurs any moral imperative to facilitate change [34]. However, normalization is not just about getting used to certain circumstances and requires more than the passage of time and repetition. Normalization is also about becoming accustomed to the particular space, language, and people involved with the social context, and developing acceptance of the situation as normal. This requires transformation of one’s thinking and the introduction of a new way of relating to people and situations [34]. In terms of managing physical health issues in everyday life, this would entail—as suggested by the findings of our research—a shared understanding of poor physical health as ‘normal’ by the involved parties. In this case, the moral ambiguity of responding to poor physical health among people who also have persistent and severe mental health issues, may become lost in the recurrence and enduring nature of issues and events and the social construction of ‘ordinary’ everyday life [34].

‘Normalization of deviance’ is another concept drawn from sociology, that can provide insight into some of the practices that were enacted in the social contexts of this research, see for example [39,40,41]. Normalization of deviance occurs when people within a specific social context become insensitive to deviant practices (Wright et al., 2021, p. 4). Such insensitivity is described as imperceptible and developing over time, for example, as people working in certain contexts continue to bend rules or reduce the standards of their work to the point that ‘a new normal’ is reached over time. An important factor in normalization of deviance is that divergent practices continue, because negative outcomes do not appear to immediately follow these actions [39,40]. The trouble begins as other critical factors line up and disasters occur [39]. Even though this concept was based on work to understand the Challenger space shuttle disaster in 1986, it is highly applicable to contemporary health care settings, where normalization of deviance has a strong, pervasive presence [39,41].

In relation to our research, the concept of ‘deviant practices’ primarily relates to the MHCPs engaged in providing care to participants with schizophrenia. The existence of deviant practices in this context can—through the beliefs, attitudes, and work practices of MHCPs described in our findings—be influential on the everyday practices of the participants with schizophrenia. Understanding the findings of our study through the lens of normalization of deviance, means the lack of active engagement in physical health-related work by MHCPs could reinforce the ‘invisibility’ of the prevailing physical health issues in people with schizophrenia. Consequently, social processes that reinforce what could be characterised as a slowly developing disaster of poor physical health and excess mortality among people with schizophrenia, are sustained through the acceptance of these issues in the context of everyday work. Banja (2010) explains how disaster is not triggered by a single mistake or a single case of bending the rules. Rather, serious harm occurs as a group of people commit multiple seemingly innocent mistakes that reduce standards of practice [41]. In this case, MHCPs being ill-equipped to recognise, respond to, and engage with participants with physical health issues, might seem relatively insignificant in a single situation. However, when understood as repeated patterns of deviant practices that result in the creation of ‘a new normal’, this lack of engagement can cause serious damage over time.

Previous research has described how management of physical health issues in the context of some mental health services entails dilemmas, which might lead to MHCPs resorting to deviant practices, such as bending clinical guidelines and omitting certain tasks, in order to safeguard fragile relationships with mental health service users [19,20]. Different factors can lead to such deviant practices. They have been described as originating in productivity pressure, generalised complacency, social pressure or negative acculturation, or peer pressure to comply with deviant practices [39]. All of these seem possible in the case of our research, where MHCPs described various factors related to the organisation of care and potential conflicts that posed barriers to the provision of physical health care. By not triggering an immediate overt negative outcome, normalization of deviance reinforces practices that are related to ‘cutting corners’ [39]. In our study, for example, MHCPs experienced pressure to comply with organisational demands to deliver mental health care as their primary task, which ‘forced’ them to de-prioritise management of potential physical health issues [26]. In highlighting these potential deviant practices, we do not suggest that the MHCPs in this ethnographic study were not doing their very best, nor do we aim to apportion blame or mis-place responsibility. Rather, we acknowledge that they were acting in a way they believed to be in the best interest of the participants with schizophrenia, based on their training and experience, and locally accepted practices. These points of discussion represent attempts to provide further explanatory ideas about what goes on in the process of managing physical health issues when understood as practices that are continuously shaped as part of everyday life interactions.

### Limitations

Ethnographic research is small-sample research. This is often advised to ensure possibility for deep immersion into the social worlds of the ones being studied [23]. Conducting ethnographic research generates a large amount of data that are typically unstructured in nature. In the study reported here, this was seen in the way that different participants contributed to different combinations of data materials. As the field notes provided the overall basis of the research material, the unstructured feature of the remaining data was not an issue of specific concern. Rather this way of adapting the methods to the individual participants seemed to strengthen the general generation of data material [23]. To establish trustworthiness of data and validation of the interpretations and conclusions made in this small-sample ethnographic study, triangulation of research methods, data sources and researcher perspectives were applied [23].

Despite the limitations of this research, it represents an example of small-sample research, in which extended contextual knowledge contributed to the construction of in-depth descriptions and interpretations of patterns of social behaviour and interactions in everyday life, that makes it possible for the reader to assess how the patterns related to managing physical health presented in the findings are relevant and potentially transferable to similar contexts of care.

## 5. Conclusions

Across the social context of everyday life of the participants with schizophrenia, the approach to manage physical health issues was characterised by a lack of recognition of the seriousness of physical health issues being a part of everyday life among people with schizophrenia. Poor physical health among people with schizophrenia was accounted for as unimportant by both MHCPs and the people experiencing physical health issues—resulting in a deadlock situation of no physical health improvements and joint acceptance of poor physical health as a “new normal”. The integration of findings contributes new knowledge about contextual factors and social interactional processes within this social context that sustained co-construction of poor physical health among people with schizophrenia as normal. At the individual level, this shared understanding contributed to participants with schizophrenia sustaining inexpedient management strategies of “modifying” behaviour or “retreating” from everyday life when experiencing physical health issues.

Additional research is needed to gain knowledge on interventions or interaction strategies that are beneficial to the management of coexisting mental health and physical health issues in everyday life settings. Such studies should aim to challenge any existing understandings of poor physical health among people with severe mental illness as normal. Furthermore, it is worth considering how engaging in the management of physical health issues could be framed as something concrete for mental health care professionals and people with severe mental illness to collaborate on—rather than something we do not do in mental health settings.

## Figures and Tables

**Table 1 ijerph-20-06133-t001:** Overview of methods, data material, and analyses.

Data Type	Participants	(n)	Method	Entities	Description of Research Activity	Data Material	Analysis	Reference
Fieldwork	Participants with schizophrenia	9	Participant observation	505 h	Researcher’s participation in everyday life activities	Field notes	Thematic	Lerbæk et al. (2021); “Modifying” or “Retreating”- Self-management of physical health among a group of people with schizophrenia. An ethnographic study from Denmark. In: *International Journal of Mental Health Nursing*; 30:6, pp. 1575–1587 [28]
Fieldwork	Participants with schizophrenia	4	Individual, semi-structured interviews	4 interviews	Semi-structured interviews informed by an interview guide, which was adjusted to fit the individual participant, based on knowledge gained through field work	Transcripts of interviews	Thematic	Ibid.
Fieldwork	Participants with schizophrenia	6	Audio-recorded conversations	27 audio-recorded conversations	Informal, unstructured conversations with participants, exploring everyday life topics	Audio-recordings (partly transcribed)	Thematic	Ibid.
Fieldwork	Participants with schizophrenia	8	PANSS ^1^ interview	8 interviews	Semi-structured interview based on standard interview guide	Transcript of interviews	Thematic	
Fieldwork	Participants with schizophrenia	8	EASE ^2^ interview	8 interviews	Semi-structured interview based on standard interview guide ^3^	Transcript of interviews	Thematic	Ibid.
Focus groups	MHCPs ^3^	22	Focus groups	3 focus groups	Focus group discussions facilitated by themes related to management of physical health issues in everyday work situations	Transcripts of focus groups	Discourse	Lerbæk et al. (2019) “Mental health care professionals' accounts of actions and responsibilities related to managing physical health among people with severe mental illness”. In: *Archives of Psychiatric Nursing*; 33, pp. 174–181 [26]
Interviews with key informants	MHCPs ^3^	5	Individual, semi-structured interviews	5 interviews	Semi-structured interviews informed by an interview guide, which was adjusted to fit the individual participant	Audio-recordings (partly transcribed)	Thematic	Lerbæk et al. (2021); Barriers and possible solutions to providing physical health care in mental health care: A qualitative study of key informants’ perspectives. *In: Issues in Mental Health Nursing*; 42:5, pp. 463–472 [27]

^1^ Positive and Negative Syndrome Scale (PANSS), see [32]. ^2^ Examination of Anomalous Self-Experience (EASE), see [33]. ^3^ Mental health care professionals is abbreviated MHCPs.

**Table 2 ijerph-20-06133-t002:** Display of integrated interpretation - examples.

Theme	Participants with Schizophrenia	Mental Health Care Professionals	Integrated Interpretation
*De-prioritizing physical health issues in everyday life and practice*	Experience different types of physical health issues that limit everyday life practices. Physical health issues are described as ‘*not of importance’*, ‘*not something to worry about*’, and ‘*just an illusion’*. Physical health issues are ‘*not a priority*’ in everyday life and ‘*not something to waste time talking about*’. Managing mental health issues is the main focus in everyday life.	Believe that people with schizophrenia have poorer physical health than the general population. Physical illness is often poorly managed in everyday life. People with schizophrenia are described as ‘*unhealthy’* and lead ‘*unhealthy lifestyles’*. Managing physical health is ‘*not a priority’* in everyday work practices. De-prioritizing physical health has become accepted practice in these settings. Physical health issues are only prioritized if something ‘*is visibly wrong’*, if MHCPs detect that someone is ‘*acting different than normal*’, or if the issue can ‘*become a danger to life’*. Some MHCPs ‘*have gotten used to this miserable health status’* and recognize that *‘we don’t really get anything done about it’*.	Prevailing physical health issues are impacting negatively on everyday lives of people with schizophrenia. Physical health issues are not recognized as important by participants with schizophrenia or MHCPs and are de-prioritized in everyday life and work practices.
*Limited management of physical health issues in everyday life settings*	Attempt to manage physical health issues in everyday life by modifying their behavior to avoid discomfort or by retreating to recover. These management strategies caused increased social isolation, affected mental health negatively and did not result in improvement of their physical health. Experienced a lack of ability to manage self-care when feeling physically ill. Repeated their typical patterns of management of physical health issues as it was ‘*the easiest*’ thing to do, even though it did not help. Risk potential worsening in physical health.	Attitudes and beliefs of MHCPs included limited expectations of peoples’ capability into managing their health due to the severe and persistent mental illness; “*things get stranded*”. Extreme examples of SMI was used by MHCPs to underline the challenges they faced if they were to engage in management physical health issues as part of everyday work practices; e.g. paranoia was ”*difficult to compete with*” and persons experiencing delusions was “*imposible to reach*”. Efforts to achieve real change in physical health among those with schizophrenia was depicted as pointless, hence justifying situations where MHCPs ended up “*doing nothing*” or “*giving up*”.	In these social settings of everyday life, management of physical health issues were very limited. During the time of the study, neither participants with schizophrenia or MHCPs took action to change the way things were. A lack of interaction and intervention from both sides constituted a silent accept of poor physical health as normal among those with schizophrenia.

## Data Availability

Not applicable.

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
