# Peer review of "“It’s Not Important”—The Social Constructing of Poor Physical Health as ‘Normal’ among People with Schizophrenia: Integrated Findings from an Ethnographic Study"

_ijerph, 2023, doi:10.3390/ijerph20126133_

Round 1

Reviewer 1 Report

INTRODUCTION - This section is concise with a clearly-defined rationale for the study (i.e., exploring physical health among people with schizophrenia - an under-researched area with a vulnerable group). Background information is germane and sufficiently referenced. It is noted that this manuscript reflects an aspect of a larger ethnographic study. No hypotheses are posed but rather an area of exploration. So, eliciting insights into how this community manage physical health issues is the aim. 

METHODS - This study used an ethnographic approach based on social construction assumptions and involved observations, interviews and focus groups with individuals diagnosed with schizophrenia and mental health practitioners. Overall, the procedure is clear and replicable. The use of thematic analysis and discursive approaches makes for a rich analytical strategy. Flow-chart of procedure is clear and helpful. The participants are not well-described, and although the reader is directed to other research, a brief summary would have been useful here (e.g., demographics, gender, SES, etc.).

RESULTS - This section is structured in a logical way. Results are clearly laid-out by theme. Each theme is defined, articulated and discussed in light of participant data, and reflects an integration of the various data strands. Pertinent quotes are included to support or illustrate a specific idea within a theme.  

DISCUSSION - This section presented a fair assessment of the findings. The Main findings (compromised physical health and poor management thereof) were integrated and expanded in terms of their significance to the wider issue of health and wellbeing of persons with schizophrenia. Study limitations are noted with reported efforts to address these to minimise impact on the trustworthiness of the data (e.g., small samples and looser structures of ethnographic work).  

CONCLUSIONS - These are clearly stated and justified by the results.

REFERENCES - Appear to be in order. 

Author Response

INTRODUCTION - This section is concise with a clearly-defined rationale for the study (i.e., exploring physical health among people with schizophrenia - an under-researched area with a vulnerable group). Background information is germane and sufficiently referenced. It is noted that this manuscript reflects an aspect of a larger ethnographic study. No hypotheses are posed but rather an area of exploration. So, eliciting insights into how this community manage physical health issues is the aim. 

Response: Thank you for this comment.

METHODS - This study used an ethnographic approach based on social construction assumptions and involved observations, interviews and focus groups with individuals diagnosed with schizophrenia and mental health practitioners. Overall, the procedure is clear and replicable. The use of thematic analysis and discursive approaches makes for a rich analytical strategy. Flow-chart of procedure is clear and helpful. The participants are not well-described, and although the reader is directed to other research, a brief summary would have been useful here (e.g., demographics, gender, SES, etc.).

Response: Thank you for this comment. We have added demographic information on the participants and still refer to how detailed information has been published elsewhere. We do not have specific SES. We also still refer to how detailed information has been published elsewhere.

RESULTS - This section is structured in a logical way. Results are clearly laid-out by theme. Each theme is defined, articulated and discussed in light of participant data, and reflects an integration of the various data strands. Pertinent quotes are included to support or illustrate a specific idea within a theme.  

Response: We are glad that the results are presented in a logic and clear manner.

DISCUSSION - This section presented a fair assessment of the findings. The Main findings (compromised physical health and poor management thereof) were integrated and expanded in terms of their significance to the wider issue of health and wellbeing of persons with schizophrenia. Study limitations are noted with reported efforts to address these to minimise impact on the trustworthiness of the data (e.g., small samples and looser structures of ethnographic work). 

Response: Thank you.

CONCLUSIONS - These are clearly stated and justified by the results.

REFERENCES - Appear to be in order. 

Response: Thank you

Reviewer 2 Report

The authors conduct an ethnographic investigation on how individuals with schizophrenia perceive physical health. This topic holds significant importance in the medical field, as Thornicroft highlighted in his influential paper from 2011. The study represents a comprehensive contribution to this crucial area of research and is effectively written. While I lack expertise in the methodology, I believe it would be beneficial to seek a specialist's input to review the paper. Specifically, I am curious about the general acceptance of the proposed reanalysis.

Author Response

The authors conduct an ethnographic investigation on how individuals with schizophrenia perceive physical health. This topic holds significant importance in the medical field, as Thornicroft highlighted in his influential paper from 2011. The study represents a comprehensive contribution to this crucial area of research and is effectively written. While I lack expertise in the methodology, I believe it would be beneficial to seek a specialist's input to review the paper. Specifically, I am curious about the general acceptance of the proposed reanalysis.

Response: Thank you for this positive recognition of our work. Other reviewers have been brought in to comment on the methododology.

Reviewer 3 Report

Dear Authors:

Congratulations on your research, you cover a sector of the population that requires timely attention.

Your paper is characterized by originality, and it is well structured, however, could be improved:

- Abstract: Add more information about the methods

- Methods: Very detailed and adequate.

- Conclusion: Add recommendations for future studies.

I identified some spelling errors in lines: 57, 58, 62, 65

I identified some spelling errors in lines: 57, 58, 62, 65

Author Response

Dear Authors:

Congratulations on your research, you cover a sector of the population that requires timely attention.

Your paper is characterized by originality, and it is well structured, however, could be improved:

- Abstract: Add more information about the methods

Response: Thank you for this suggestion. We have added information om method as suggested.

- Methods: Very detailed and adequate.

Response: Thank you

- Conclusion: Add recommendations for future studies.

Response: This is an important aspect and we have added perspectives for future research to the conclusion.

I identified some spelling errors in lines: 57, 58, 62, 65

Response: Thank you for pointing out these errors. We have corrected them.

Reviewer 4 Report

It was my pleasure to read this paper which I suggest to raise very important questions for psychiatric practice and the society in general. I think the article is of a high academic quality and may be published as it is. But I also have minor recommendations:

line 56: it is hard to understand what is meant by "structural level", in what sense. If I got the idea well, you mean organizational level here. Maybe it would be better to reformulate

Materials: I understood that the data has been published in a series of article, but still here, in this very text, it seems to be lack of description of the methods. For instance, I would suggest explaining a bit more about 1) purposeful sampling, 2) where the participants lived (it is written in a very short and culturally specific form which may mean a little for a foreigner), 3) how the information was gathered (in what kind of activities and informal talks? what kind of questions the interviews or focus groups contained? - at least some examples)

Table 2 would rather be better in the Results section

The last row in the table 2, the second column - I think there are typos in "was/were"

Line 159 - I would remove the word "new" themes. I think the idea is that they are new relative to the previously published results. But here they are not new, they are the only and the main ones.

Discussion - I have just one comment: you showed very well that medical staff do not believe that their efforts to manage their patients' physical illness may be effective due to their mental state. This is related to the problems of contact, cooperation, therapeutical allience, adherence to treatment etc. So, the idea is that if we can't manage well enough their mental state how could we manage even more? But on the other hand, and to the opposite, physical ill health may be regarded as a potentially better, easier point to collaborate on. I mean that this content may be re-assessed as a good possibility to maintain a reasonable and effective interaction with a patient, which may become then a starting point to interact better on his mental issues. I mean it is much easier to convince a patient that he has some physical issues (when it indeed hurts) than to convince him that he has delusions etc. I hope these ideas may be useful for the discussion, because this is about concrete practical applications of the data you provide. 

I'm very inspired by your work and I'm looking forward to see it published to recommend to my colleagus.

Author Response

It was my pleasure to read this paper which I suggest to raise very important questions for psychiatric practice and the society in general. I think the article is of a high academic quality and may be published as it is. But I also have minor recommendations:

line 56: it is hard to understand what is meant by "structural level", in what sense. If I got the idea well, you mean organizational level here. Maybe it would be better to reformulate

Response: Thank you for pointing out this lack of clarity. The concept ‘structural level’ has been clarified with some examples. The examples are drawn from the work of Melamed et al. (2019) who identified with barriers at micro, meso, and macro levels. The ‘structural level’ referred to in our text, equates with macro-level barriers (for example, geographical distance to services, conflicting health care appointments, financial constraints). We hope that the additions clarify the text.

Materials: I understood that the data has been published in a series of article, but still here, in this very text, it seems to be lack of description of the methods. For instance, I would suggest explaining a bit more about 1) purposeful sampling, 2) where the participants lived (it is written in a very short and culturally specific form which may mean a little for a foreigner), 3) how the information was gathered (in what kind of activities and informal talks? what kind of questions the interviews or focus groups contained? - at least some examples)

Response: You point to some important aspects, and we have added description of the purposeful sampling strategy used in the study. We have also added text to elaborate on where the participants with schizophrenia lived.

We have not elaborated further on the data generation methods in text, as these are described in Table 1 ‘Overview of data materials and analysis’. We have added ‘methods’ to the heading of the table to highlight that it also includes methods.

Table 2 would rather be better in the Results section

Response: Table 2 is presented the first time in relation to the description of the process of analysis in order to provide the reader with a form of ‘map’ linking analysis and findings. A second objective is to present a short and clear example of the analysis and the contributions of the different perspectives. We value the suggestion to move the table to the findings section but have decided to keep it in the current location of the analytical strategy section so as to keep the abovementioned functions. However, we have added a reference to the table in the initial text of the results section.

The last row in the table 2, the second column - I think there are typos in "was/were"

Response: Thank you for pointing this out. We have corrected this error.

Line 159 - I would remove the word "new" themes. I think the idea is that they are new relative to the previously published results. But here they are not new, they are the only and the main ones.

Response: We have removed the word as suggested.

Discussion - I have just one comment: you showed very well that medical staff do not believe that their efforts to manage their patients' physical illness may be effective due to their mental state. This is related to the problems of contact, cooperation, therapeutical allience, adherence to treatment etc. So, the idea is that if we can't manage well enough their mental state how could we manage even more? But on the other hand, and to the opposite, physical ill health may be regarded as a potentially better, easier point to collaborate on. I mean that this content may be re-assessed as a good possibility to maintain a reasonable and effective interaction with a patient, which may become then a starting point to interact better on his mental issues. I mean it is much easier to convince a patient that he has some physical issues (when it indeed hurts) than to convince him that he has delusions etc. I hope these ideas may be useful for the discussion, because this is about concrete practical applications of the data you provide. 

Response: Thank you for the reflections on how physical health issues may be easier to collaborate on than mental health issues. This perspective is one to consider moving forward with research in this area. However, in the study reported here, the perspectives and behaviours of participants were rather opposite; that if they cannot manage to stabilise the mental health state first, then they would not even try to manage the physical health issues.

We value the reviewer’s reflections but have respectfully decided not to include them in the current discussion. They will however serve as inspiration for future studies, and we have included this in the conclusion.

I'm very inspired by your work and I'm looking forward to see it published to recommend to my colleagus.

Response: Thank you for this recognition of our work.